# A Note On The Stability Of The Focal Loss

**Martijn P. van Leeuwen** *m.p.vanleeuwen@tilburguniversity.edu*
*Dept. of Intelligent Systems, Research Center for Cognitive Science and Artificial Intelligence,*
*Tilburg School of Humanities and Digital Sciences, Tilburg University*

**Koen V. Haak** *k.v.haak@tilburguniversity.edu*
*Dept. of Intelligent Systems, Research Center for Cognitive Science and Artificial Intelligence,*
*Tilburg School of Humanities and Digital Sciences, Tilburg University*

**Görkem Saygili** *g.saygili@tilburguniversity.edu*
*Dept. of Intelligent Systems, Research Center for Cognitive Science and Artificial Intelligence,*
*Tilburg School of Humanities and Digital Sciences, Tilburg University*

**Eric Postma** *e.o.postma@tilburguniversity.edu*
*Dept. of Computational Cognitive Science, Research Center for Cognitive Science and Artificial Intelligence,*
*Tilburg School of Humanities and Digital Sciences, Tilburg University*

**L.L. Sharon Ong** *l.l.ong@tilburguniversity.edu*
*Dept. of Intelligent Systems, Research Center for Cognitive Science and Artificial Intelligence,*
*Tilburg School of Humanities and Digital Sciences, Tilburg University*

**Reviewed on OpenReview:** *https://openreview.net/forum?id=eCYActnGbu*

## Abstract

The Focal Loss is a widely deployed loss function that is used to train various types of deep learning models. It is a modification of the cross-entropy loss designed to mitigate the effect of class imbalance in dense object detection tasks. By downweighting the losses for easy, correctly classified samples, the method places more emphasis on harder, misclassified ones. As a result, gradient updates are not dominated by samples that the model already handles correctly. The downweighting of the loss is achieved by scaling the cross-entropy loss with a term that depends on a focusing parameter $\gamma$. In this paper, we highlight an unaddressed numerical instability of the Focal Loss that arises when this focusing parameter is set to a value between 0 and 1. We present the theoretical basis of this numerical instability, show that it can be detected in the computation of Focal Loss gradients, and demonstrate its effects across several classification and segmentation tasks. Additionally, we propose a straightforward modification to the original Focal Loss to ensure stability whenever these unstable focusing parameter values are used.

## 1 Introduction

The Focal Loss is widely used in one-stage object detectors, medical imaging, image segmentation, and pose estimation tasks. (Terven et al., 2023). This function modifies the distribution-based cross-entropy loss by introducing a focusing parameter ($\gamma$) that downweights the penalty that is applied to "easy" examples (Lin et al., 2017). This downweighting of the loss prevents gradients from being dominated by easy examples, allowing the model to focus on difficult samples. (Lin et al., 2017). This is particularly useful when the training data contains a large proportion of background (or other) class samples.

The selection of parameter $\gamma$, and with this, the extent to which easy examples are downweighted, should be done via cross-validation (Lin et al., 2017). Lin et al. (2017) reported that using a $\gamma$ of 2 led to the

best results in their experiments (Lin et al., 2017). There is, however, a limit to what $\gamma$ values can be used. Selecting $\gamma$ values much larger than 2 has been shown to result in gradients close to 0 for relatively low model outputs, causing training to fail (Mukhoti et al., 2020). This paper will shed light on the other end of the spectrum, showing that the Focal Loss gradients can become unstable whenever $\gamma$ is too small. More specifically, we address a numerical instability of the Focal Loss that arises whenever a $\gamma$ is set to a value between 0 and 1. These $\gamma$ values can cause the Focal Loss derivative to become undefined and destabilize model training due to a singularity. This singularity arises whenever the Focal Loss, in combination with a $\gamma$ on the open unit interval, is used to learn a task for which a model can confidently predict a correct class label. We will demonstrate that this numerical instability of the Focal Loss is not only a theoretical problem by showing that a simple convolutional neural network (CNN), a vision transformer (ViT) (Wu et al., 2020), and a 2D U-net (Ronneberger et al., 2015) can return undefined loss values during training whenever $\gamma$ values on the open unit interval are used. Henceforth, we will refer to the $\gamma$ values between 0 and 1 as unstable $\gamma$ values.

The original Focal Loss paper (Lin et al., 2017) did not address this numerical instability and presents experimental results generated with $\gamma$ between 0 and 1. While their findings show that training with unstable $\gamma$ values does not always lead to instability, our experiments highlight certain training conditions that increase the chance of encountering this instability.

For instance, will show that using unbalanced datasets increases the likelihood of encountering numerical instability when using unstable $\gamma$ values. Especially in the medical field, collected datasets are often class-imbalanced as a result of the prevalence of diseased patients (Salmi et al., 2024). Given that the Focal Loss is designed to deal with class imbalance, using the Focal Loss in medical machine learning tasks is not an uncommon approach (Ahmed et al., 2022; Romdhane & Pr, 2020; Tran et al., 2019). Class imbalance is also encountered in other fields such as autonomous driving (Chen & Qin, 2022) and fault detection (Zareapoor et al., 2021). Due to the class imbalance in these tasks, the Focal Loss is also employed in research on these topics (Wei et al., 2022; Carranza-García et al., 2021). The optimal $\gamma$ value for any of these applications could be in the unstable range. To avoid numerical instability, it may be necessary to use a suboptimal value, which can in turn limit overall model performance. It is therefore crucial to address this instability, so that all possible $\gamma$ can be considered without risking training instability.

This instability is likely not confined to the Focal Loss itself but may also influence other loss functions that build upon or are drived from it. In recent years, various modifications of the Focal Loss have been proposed, such as the Generalized Focal Loss (Li et al., 2020), the Adaptive Focal Loss (Islam et al., 2024), and the Unified Focal Loss (Yeung et al., 2022). The Generalized Focal Loss extends the Focal Loss to jointly model classification confidence and localization quality in a joint representation, applying a focusing parameter similar to the original Focal Loss. In the Adaptive Focal Loss (Islam et al., 2024), the focusing parameter is dynamically adjusted during training, and the Unified Focal Loss combines the Focal Loss with the Focal Tversky Loss (Abraham & Khan, 2019) but constrains the focusing parameter to a fixed range between 0 and 1. All these new loss functions build upon the Focal Loss and similarly employ the focusing parameter as in the original formulation. As a result, they are likely to exhibit the same numerical instability that we address in this paper.

In this paper, we highlight the origin of the instability and demonstrate how to resolve it. Without altering the original behavior of the Focal Loss, our proposed stabilization approach ensures compatibility with existing methods that employ the Focal Loss using these unstable $\gamma$ values. Our contribution is therefore not only to provide evidence that the instability exists, but also to apply a minimally invasive modification to eliminate it, while simultaneously eliminating the instability in all Focal Loss variants that use the focusing parameter similarly.

## 2 Methods

This section will review the definition of the Focal Loss and its derivative to explain the origin of the instability. We then show that the numerical instability becomes apparent when computing the gradients of the original Focal Loss with unstable $\gamma$ values. Additionally, to show that this instability is not only a

theoretical obstacle, we will demonstrate that under certain conditions, the instability can be induced in binary classification and 2D segmentation tasks. Lastly, we present a modification of the original Focal Loss that eliminates the instability whenever unstable $\gamma$ values are used.

## 2.1 Cross-entropy and the Focal Loss

The Focal Loss was introduced to address class imbalance by reducing the effect of easily classifiable examples, thereby placing more emphasis on harder, misclassified ones (Lin et al., 2017). This is achieved by modulating the standard cross-entropy loss in Equation (1) with a scaling factor that is based on the prediction error and a focusing parameter $\gamma$. This modulating factor ensures that the smaller the prediction error becomes, the more the cross-entropy is downscaled. In other words, predictions that are closer to the correct label (easy examples) are downscaled by the focusing parameter $\gamma$. Note that whenever a $\gamma$ of 0 is used, the Focal Loss simplifies to the cross-entropy loss. For simplicity, without loss of generality, we will simplify the cross-entropy loss function to the binary cross-entropy loss and reformulate the equations to a foreground ($\mathcal{L}_{fg}$) and background loss ($\mathcal{L}_{bg}$). For consistency, we make use of the same notation for the ground truth ($y$) and model output ($p$) as was used in the original Focal Loss paper (Lin et al., 2017). While the original Focal Loss paper reformulates the loss as a foreground loss for notational convenience, this paper explicitly highlights both the foreground and background components of the Focal Loss.

$$\mathcal{L}_{\mathrm{CE}}(y, p) = - \underbrace{y \log(p)}_{\mathcal{L}_{fg}} - \underbrace{(1 - y) \log(1 - p)}_{\mathcal{L}_{bg}} \tag{1}$$

$$\mathcal{L}_{\mathrm{F}}(y, p, \gamma, \alpha_t) = \underbrace{-\alpha_t y \, (1 - p)^\gamma \log(p)}_{\mathcal{L}_{fg}} - \underbrace{(1 - \alpha_t)(1 - y) \, p^\gamma \log(1 - p)}_{\mathcal{L}_{bg}} \tag{2}$$

The Focal Loss, as defined in Equation (2), downscales both the foreground and background loss equally with the focusing parameter $\gamma$. As shown in Equation 2, the Focal Loss also includes a parameter $\alpha_t$ that can be used to scale the contribution of the foreground and the background loss relative to each other.

## 2.2 Derivative of the Focal Loss

Figure 1a and Figure 1b show the foreground and background components of the Focal Loss for different model outputs $p$ when changing the value for the focusing parameter $\gamma$. These plots show that an increase in $\gamma$ will cause the Focal Loss to show near-zero loss values for model outputs close to the ground truth label, consequently lowering their associated gradients. As we previously decomposed the Focal Loss into

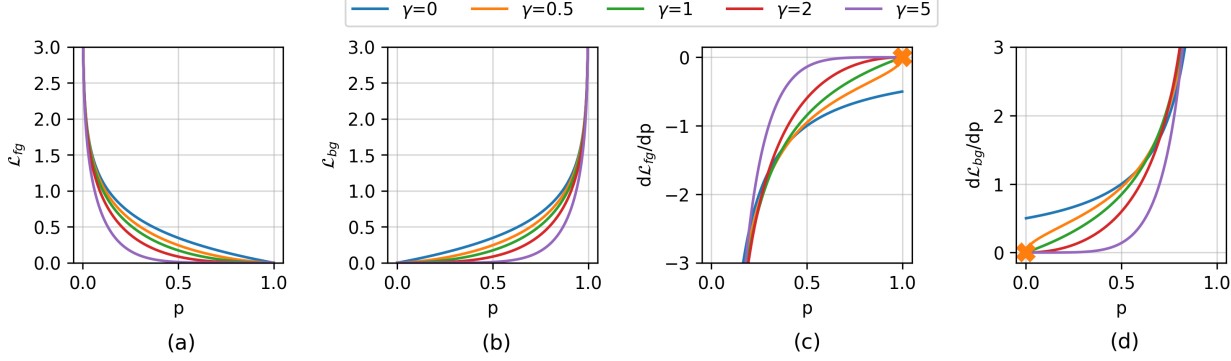

Figure 1: Foreground ($a$) and background ($b$) components of the Focal Loss and their associated foreground ($c$) and background ($d$) derivatives. The losses and derivatives were calculated with different $\gamma$ values and a fixed $\alpha_t$ of 0.5. The orange markers indicate undefined gradient values as a result of a division by 0 in the derivative of the Focal Loss and represent the model outputs when $p$ is equal to 1 in Figure ($c$) and 0 in Figure ($d$).

a foreground and background loss, we can define the Focal Loss derivative as the sum of the foreground and background components, as shown in Equation (3). These two components are defined in Equations (4) and (5), and are displayed in Figure 1c and Figure 1d. The derivation of these equations is included in Appendix A.1.

$$\frac{d\mathcal{L}_F(p, \gamma, \alpha_t)}{dp} = \frac{d\mathcal{L}_{fg}(p, \gamma, \alpha_t)}{dp} + \frac{d\mathcal{L}_{bg}(p, \gamma, \alpha_t)}{dp} \tag{3}$$

$$\frac{d\mathcal{L}_{fg}(p, \gamma, \alpha_t)}{dp} = \alpha_t \left( \gamma(1-p)^{\gamma-1} \log(p) - \frac{(1-p)^\gamma}{p} \right) \tag{4}$$

$$\frac{d\mathcal{L}_{bg}(p, \gamma, \alpha_t)}{dp} = -(1-\alpha_t) \left( \gamma\, p^{\gamma-1} \log(1-p) - \frac{p^\gamma}{1-p} \right) \tag{5}$$

## 2.3 Focal Loss Instability

In the derivative of the Focal Loss, a $(\gamma - 1)$ exponent is introduced in both the foreground and background components. Whenever $0 < \gamma < 1$, this $(\gamma - 1)$ term becomes negative, and the model output is raised to the power of a negative number, creating a fraction with the model output in the denominator. An example of this is illustrated in Equations (6) and (7), showing the derivatives of the foreground and background loss for $\gamma = 0.5$.

$$\frac{d\mathcal{L}_{fg}(p, \alpha_t)}{dp}\Big|_{\gamma=0.5} = \alpha_t \left( \frac{0.5}{\sqrt{1-p}} \log(p) - \frac{\sqrt{1-p}}{p} \right) \tag{6}$$

$$\frac{d\mathcal{L}_{bg}(p, \alpha_t)}{dp}\Big|_{\gamma=0.5} = -(1-\alpha_t) \left( \frac{0.5}{\sqrt{p}} \log(1-p) - \frac{\sqrt{p}}{1-p} \right) \tag{7}$$

The scenarios involving a $\gamma$ value of 0.5, where $p$ equals 1 for the foreground and 0 for the background component, are shown in Equations (8) and (9). These equations show that at these points, the fraction introduced in the derivative leads to a division by 0, creating a singularity that causes training instability. Although in this example $\gamma$ was set to 0.5, this singularity holds for all $\gamma$ values between 0 and 1, as all these values introduce a fraction in the Focal Loss derivative with the model output in the denominator.

$$\frac{d\mathcal{L}_{fg}(\alpha_t)}{dp}\Big|_{p=1,\gamma=0.5} = \alpha_t \left( \frac{0}{\sqrt{0}} - 0 \right) = \text{Undefinded} \tag{8}$$

$$\frac{d\mathcal{L}_{bg}(\alpha_t)}{dp}\Big|_{p=0,\gamma=0.5} = -(1-\alpha_t) \left( \frac{0}{\sqrt{0}} - 0 \right) = \text{Undefinded} \tag{9}$$

Consider training a binary classification model with the Focal Loss and a $\gamma$ value of 0.5, to distinguish a foreground class from a background class. When the foreground is easily separated from the background class, the model will quickly produce high model outputs for the correct classes. Whenever these model outputs are equal (within floating-point precision) to the ground truth value ($y = p$), the derivative will become undefined due to a division by 0, consequently triggering the instability. When faced with a more challenging task, the model is unlikely to produce model outputs equal to the ground truth, thereby preventing instability from being triggered. Since most deep learning tasks are complex, this is presumably why the instability is not always an issue and why it has not yet been addressed in the literature.

One important note to consider when discussing the stability of the Focal Loss is that whenever the opposite of the ground truth is predicted by the model, a $\log(0)$ is introduced in the equation of the Focal Loss, which also causes instabilities. Note that the instability that this paper addresses occurs whenever the model produces output values that are equal to the ground truth. In other words, the $\log(0)$ instability occurs when the prediction error becomes extremely large, whereas the instability that we address occurs whenever the prediction error becomes near-zero. Since machine learning models are trained to minimize prediction error, they are optimized towards a state where this instability will eventually occur if it is not addressed.

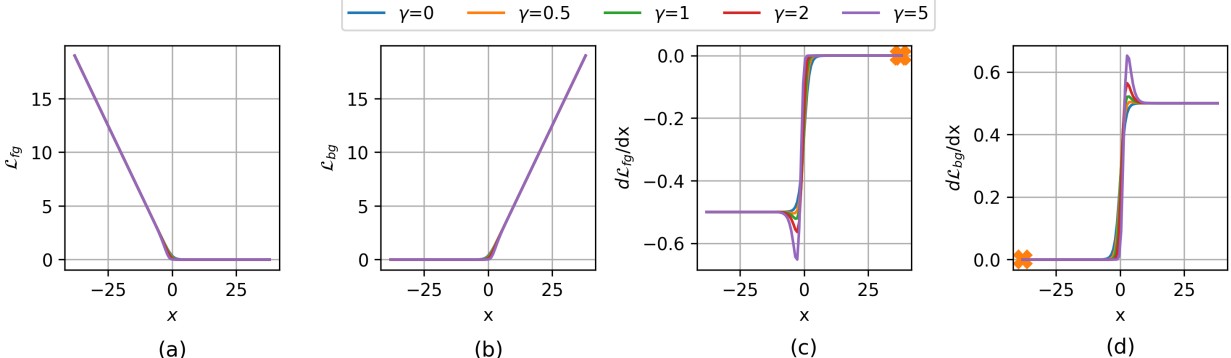

Figure 2: Computed values for the foreground (*a*) and background (*b*) components of the Focal Loss in combination with the foreground (*c*) and background (*d*) gradients. These losses and gradients were computed with model outputs (*x*) that were not yet processed by the sigmoid and ranged between [-38,38]. Both the losses and gradients were computed with the *torchvision.ops.sigmoid_focal_loss* function with an $\alpha_t$ of 0.5, and different values for $\gamma$. The orange markers indicate the values for *x* that cause "NaN" values when computing the loss gradients. They represents the model output when *x* is equal to 1 in Figure (*c*) and 0 in Figure (*d*).

### 2.3.1  Numerical Gradient Computation

In this subsection, we show that the instability occurs when computing the gradients with the original Focal Loss function (*torchvision.ops.sigmoid_ focal_loss*) used in the seminal work (Lin et al., 2017). This implementation of the Focal Loss applies a sigmoid activation function to the model outputs as shown in Equation (10), in which the model output prior to applying the sigmoid is defined as *x*. After the sigmoid, the Focal Loss is calculated using Equation (2). Figure 2a and Figure 2b show the foreground and background components of the Focal Loss when computing them with the original Focal Loss function.

$$p = \sigma(x) = \frac{1}{1 + e^{-x}} \tag{10}$$

Similar to Figure 1, Figure 2 shows the foreground and background loss in combination with their associated gradients. Figure 1 shows the Focal Loss for model outputs that are processed by a sigmoid, and Figure 2 shows the Focal Loss for unprocessed model outputs (*x*). These unprocessed output values are not confined to a range of [0,1] but can theoretically span from $[-\infty,\infty]$. Figure 2c and Figure 2d present the computed Focal Loss gradients for the foreground and background components of the Focal Loss. The orange markers in Figure 2 indicate the output values that cause the gradient to become undefined, consequently returning a "NaN". The points at which the loss becomes undefined indicate where instability arises in the computation of the original Focal Loss.

### 2.4  Stabilized Focal Loss

As described in the previous section, the numerical instability of the Focal Loss arises as a result of a division by 0 in its derivative. One commonly used approach to prevent a division by 0 in a loss function is the introduction of a smoothing constant $\epsilon$ in the denominator of the loss, which is a stabilization method that is also applied to the well-known Dice loss (Milletari et al., 2016; Sudre et al., 2017). We propose a modification of the original Focal Loss that leads to the introduction of a smoothing constant in the denominator of its derivative when unstable $\gamma$ values are used. This slightly differs from what is done to stabilize the Dice loss, in which the division by zero is prevented in the loss itself. Instead, we modify the original Focal Loss with a parameter $\epsilon$, so that the $\epsilon$ term is placed in the denominator of its derivative and not in the loss itself. This ultimetaly prevents division by zero when computing the gradient with unstable $\gamma$ values.

The modified Focal Loss and the derivatives for its foreground and background components are de-

fined as shown in Equations (11), (12), and (13). By introducing a constant denominator term independent of the model output, the proposed modification ensures numerical stability for $\gamma$ values between 0 and 1. This prevents division by zero when the prediction approaches the ground truth within floating-point precision, effectively eliminating the instability in the original Focal Loss. Note that smaller $\gamma$ values will cause the denominator to approach 0 more quickly for model outputs close to the ground truth, compared to when a larger value for $\gamma$ is used. This means that a larger $\epsilon$ is required to ensure stability whenever a smaller $\gamma$ value is used. We ran the experiments in this paper with a value of $\epsilon$ equal to $1e-3$, as it stabilized model training whenever a $\gamma$ as small as 0.1 was used.

$$\mathcal{L}_{\text{Fm}}(y, p, \gamma, \alpha_t, \epsilon) = \underbrace{-\alpha_t y \left(1 - p + \epsilon\right)^{\gamma} \log(p)}_{\mathcal{L}_{fg}} - \underbrace{(1 - \alpha_t)(1 - y)\left(p + \epsilon\right)^{\gamma}_i \log(1 - p)}_{\mathcal{L}_{bg}} \tag{11}$$

$$\frac{d\mathcal{L}_{fg}(p, \gamma, \alpha_t, \epsilon)}{dp} = \alpha_t \left( \gamma(1 - p + \epsilon)^{\gamma-1} \log(p) - \frac{(1 - p + \epsilon)^{\gamma}}{p} \right) \tag{12}$$

$$\frac{d\mathcal{L}_{bg}(p, \gamma, \alpha_t, \epsilon)}{dp} = -(1 - \alpha_t) \left( \gamma\, (p + \epsilon)^{\gamma-1} \log(1 - p) - \frac{(p + \epsilon)^{\gamma}}{1 - p} \right) \tag{13}$$

When revisiting the example in which $\gamma$ is equal to 0.5, the derivatives for the foreground and background loss become equal to Equation (14) and (15). When the model output is again equal to 1 in the foreground and 0 in the background loss, as shown in Equation (16) and (17), the division by zero is prevented by the smoothing constant $\epsilon$. The implementation details of the modified version of the Focal Loss can be found in Appendix A.3.

$$\frac{d\mathcal{L}_{fg}(p, \alpha_t, \epsilon)}{dp}\Big|_{\gamma=0.5} = \alpha_t \left( \frac{0.5}{\sqrt{1 - p + \epsilon}} \log(p) - \frac{\sqrt{1 - p + \epsilon}}{p} \right) \tag{14}$$

$$\frac{d\mathcal{L}_{bg}(p, \alpha_t, \epsilon)}{dp}\Big|_{\gamma=0.5} = -(1 - \alpha_t) \left( \frac{0.5}{\sqrt{p + \epsilon}} \log(1 - p) - \frac{\sqrt{p + \epsilon}}{1 - p} \right) \tag{15}$$

$$\frac{d\mathcal{L}_{fg}(\alpha_t, \epsilon)}{dp}\Big|_{p=1\,\gamma=0.5} = \alpha_t \left( \frac{0}{\sqrt{\epsilon}} - \frac{\sqrt{\epsilon}}{1} \right) = -\alpha_t \sqrt{\epsilon} \tag{16}$$

$$\frac{d\mathcal{L}_{bg}(\alpha_t, \epsilon)}{dp}\Big|_{p=0,\gamma=0.5} = -(1 - \alpha_t) \left( \frac{0}{\sqrt{\epsilon}} - \frac{\sqrt{\epsilon}}{1} \right) = (1 - \alpha_t)\sqrt{\epsilon} \tag{17}$$

## 2.5 Experiments

To demonstrate the instability of the Focal Loss, we conducted several experiments. First, we tested whether the instability could occur when training a basic convolutional neural network (CNN) (shown in Appendix A.4) to perform a binary classification task on the MNIST dataset (Deng, 2012). We then examined if this instability could be induced on a larger and more complex dataset, the CIFAR-10 dataset (Krizhevsky et al., 2009), using a CNN and a Vision Transformer (ViT) (Wu et al., 2020). In a final experiment, we evaluated whether the instability could be observed during the training of a 2D U-Net (Ronneberger et al., 2015) (implementation shown in A.6) on a segmentation task using the MNIST dataset. This subsection will describe how these experiments were set up.

### 2.5.1 Binary Classification - MNIST

In the first experiment, we divided the MNIST dataset (Deng, 2012), a dataset composed of handwritten numbers ranging from 0 to 9, into two classes, where a threshold determined which numbers belonged to which class. The goal of the CNN was to learn to distinguish between the two classes by training for 100 epochs. As we are interested in the stability of the Focal Loss during training, we did not focus on model performance, but rather on whether the model could complete all 100 epochs without encountering any instabilities. In this experiment, we tested stable (0,1,2,3,4,5) and unstable $\gamma$ values ($0 < \gamma < 1$) to verify that the instability only occurs when using unstable $\gamma$ values.

To transform the multiclass MNIST dataset into a dataset that could be used for a binary classifica-

Table 1: The number of samples in each class of the MNIST dataset (Deng, 2012) when using different class distributions. The class distribution was determined by a binarization threshold ranging from 0 to 8. The threshold value indicates the cut-off value for the classes belonging to class A or class B. All values larger than the threshold belonged to class B, and all classes below or equal to the threshold belonged to class A.

| Threshold | Samples Class A | Samples Class B | Class A/B ratio |
|---|---|---|---|
| 0 | 5.923 | 54.077 | 0.11 |
| 1 | 12.665 | 47.335 | 0.27 |
| 2 | 18.623 | 41.377 | 0.45 |
| 3 | 24.754 | 35.246 | 0.70 |
| 4 | 30.596 | 29.404 | 1.04 |
| 5 | 36.017 | 23.983 | 1.50 |
| 6 | 41.935 | 18.065 | 2.32 |
| 7 | 48.200 | 11.800 | 4.08 |
| 8 | 54.051 | 5.949 | 9.09 |

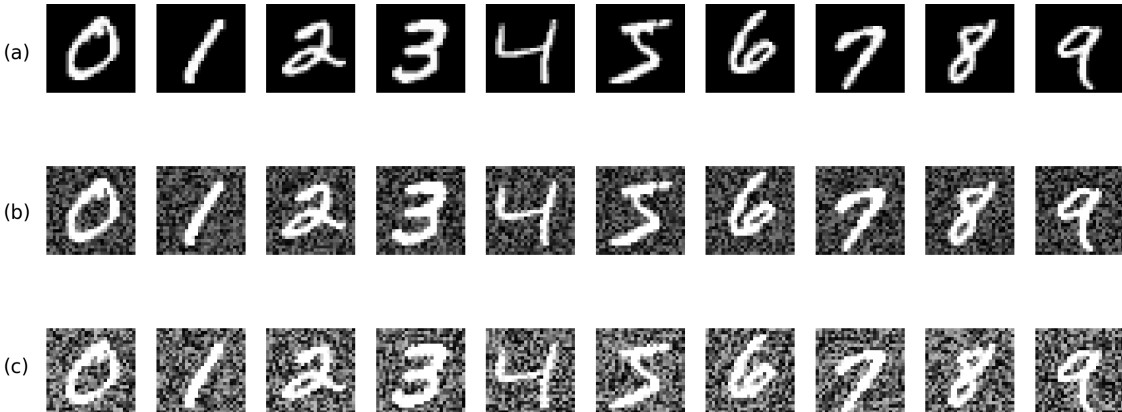

Figure 3: Example input images from the MNIST dataset (Deng, 2012) for digits 0–9. The original images are shown in (*a*), while (*b*) and (*c*) illustrate the effects of adding "Medium Noise" and "High Noise," respectively. The "Medium Noise" was generated by multiplying uniformly sampled noise between 0 and 1 by 0.5, while the "High Noise" was generated by multiplying this noise by 0.75.

tion task, we applied a threshold to the MNIST class labels, restructuring the dataset into a dataset with two classes: a foreground (A) and background (B) class. Because the number of samples for each class of the MNIST training dataset is approximately the same (Hamidi & Borji, 2010), changing this threshold allows for incrementally modifying the foreground-background ratio. Assessing the effect of changing this threshold provides insight into whether an imbalance in classes influences training stability whenever unstable $\gamma$ values are used. For example, a threshold of 4 means that the MNIST numbers with classes 0-4 belong to class A and the classes 5-9 belong to class B. Adjusting this threshold, therefore, changes the degree of class imbalance. By increasing the imbalance, we intentionally introduced a bias toward the majority class, resulting in more confident predictions for the correct labels, thereby aiming to induce instability.

The initial part of this experiment was designed to test whether increasing class imbalance in the classification task influenced the expression of the instability. The second part of the experiment repeated the initial experiment, but trained the CNN with random noise added to the input images. By adding noise, we can evaluate whether increasing the classification task's difficulty mitigates the unstable behavior of the Focal Loss. The noise was generated by sampling random values from a uniform distribution over [0, 1). These values were then scaled by 0.5 and 0.75 to produce what we refer to as "Medium Noise" and "High Noise," respectively. This randomly sampled noise was then added to the input images. Examples of input images with and without added noise are shown in Figure 3.

### 2.5.2 Binary Classification - CIFAR-10

The CIFAR-10 dataset (Krizhevsky et al., 2009) is composed of images belonging to one of the following classes: airplane, automobile, bird, cat, deer, dog, frog, horse, ship, or truck. To perform binary classification on this dataset, the classes were partitioned into two classes: animals and vehicles. Each class of the original CIFAR-10 classes contained 5000 images, and with 6 animal and 4 vehicle classes, the classes for the restructured dataset became slightly imbalanced (6:4 ratio). This setup, with its slight imbalance and the use of Focal Loss, reflects a realistic scenario commonly encountered in real-world datasets. Following dataset preparation, a Vision Transformer (ViT) (Wu et al., 2020) and a CNN were trained to perform the binary classification task. The CNN was the same model that was used in the MNIST experiment, with some minor adjustments to account for the difference in input sizes. The used ViT (Wu et al., 2020) took $224 \times 224$ images as input and divided each image into $16 \times 16$ patches, before flattening them, and projecting each patch into a learnable embedding. More information about the used ViT can be found in Appendix A.5. Both models were trained for 1000 epochs with a batch size of 128 and a $\gamma$ and $\alpha_t$ of 0.5 without using pre-trained weights. We trained on the complete training dataset, as we did not perform any hyperparameter optimization or testing.

### 2.5.3 2D Segmentation - MNIST

In the final experiment, we tested whether the instability could be induced when training a model to segment the numbers in the MNIST dataset using a 2D U-net Ronneberger et al. (2015). Since the MNIST dataset is intended for classification tasks, it does not include segmentation masks. We therefore applied a threshold of 0.5 to the input images, setting all pixels that exceeded this value to the foreground of the segmentation mask, and setting all pixels below this value to the background class. Similar to the binary classification task, we repeated the experiment after adding noise to the input data to test whether increasing the difficulty of the segmentation task influenced training stability. The noise was added after creating the segmentation masks to maintain a consistent segmentation mask across experiments. After preprocessing of the data, the 2D U-net was trained with the Focal Loss using a $\gamma$ and $\alpha_t$ of 0.5 for 1000 epochs. More details on the U-Net architecture can be found in Appendix A.6.

## 3 Results

In this section, we demonstrate the Focal Loss instability that is caused by unstable $\gamma$ values. We report our findings for the original Focal Loss and also analyze how our modified version impacts training stability.

### 3.1 Binary Classification - MNIST

For the binary classification task, we trained the CNN for 100 epochs with $\gamma$ values ranging from 0 to 5 using the different class A/B ratios that are shown in Table 1. Whenever a "NaN" was encountered during training, training was stopped, otherwise, training would continue until all 100 epochs were completed. The results for these experiments are presented in Figure 4, where the number of completed epochs is shown for different $\gamma$ and A/B class ratios. The left plot in Figure 4a shows that when $\gamma$ was set to 0 or to values greater than 1, training remained stable across all class A/B ratios, completing all 100 epochs without any signs of instability.

The middle-column figure shows the results of training with $\gamma$ values ranging from 0.1 to 0.9 in increments of 0.1 From this figure, we see that using $\gamma$ values between 0 and 1 frequently causes instability. These instabilities quickly arise, especially for smaller $\gamma$ values and unbalanced datasets. However, when using a $\gamma$ of 0.9 in combination with a relatively balanced dataset, all 100 epochs were completed. It is, however, not unlikely that whenever these models were trained for more than 100 epochs, the instability would still have been found in a later epoch.

Figure 4b and 4c show that adding noise has a mitigating effect on how quickly the instabilities are detected. These figures indicate that higher levels of noise result in a greater number of $\gamma$ values exhibiting stable behavior. Moreover, introducing noise enables a wider range of class distributions to achieve stable

training outcomes, particularly for larger $\gamma$ values. Furthermore, the left plot in Figure 4b and Figure 4c also show that adding noise did not affect the number of completed epochs when training with stable $\gamma$ values.

After stabilizing the Focal Loss with the smoothing constant, all experiments were repeated, for

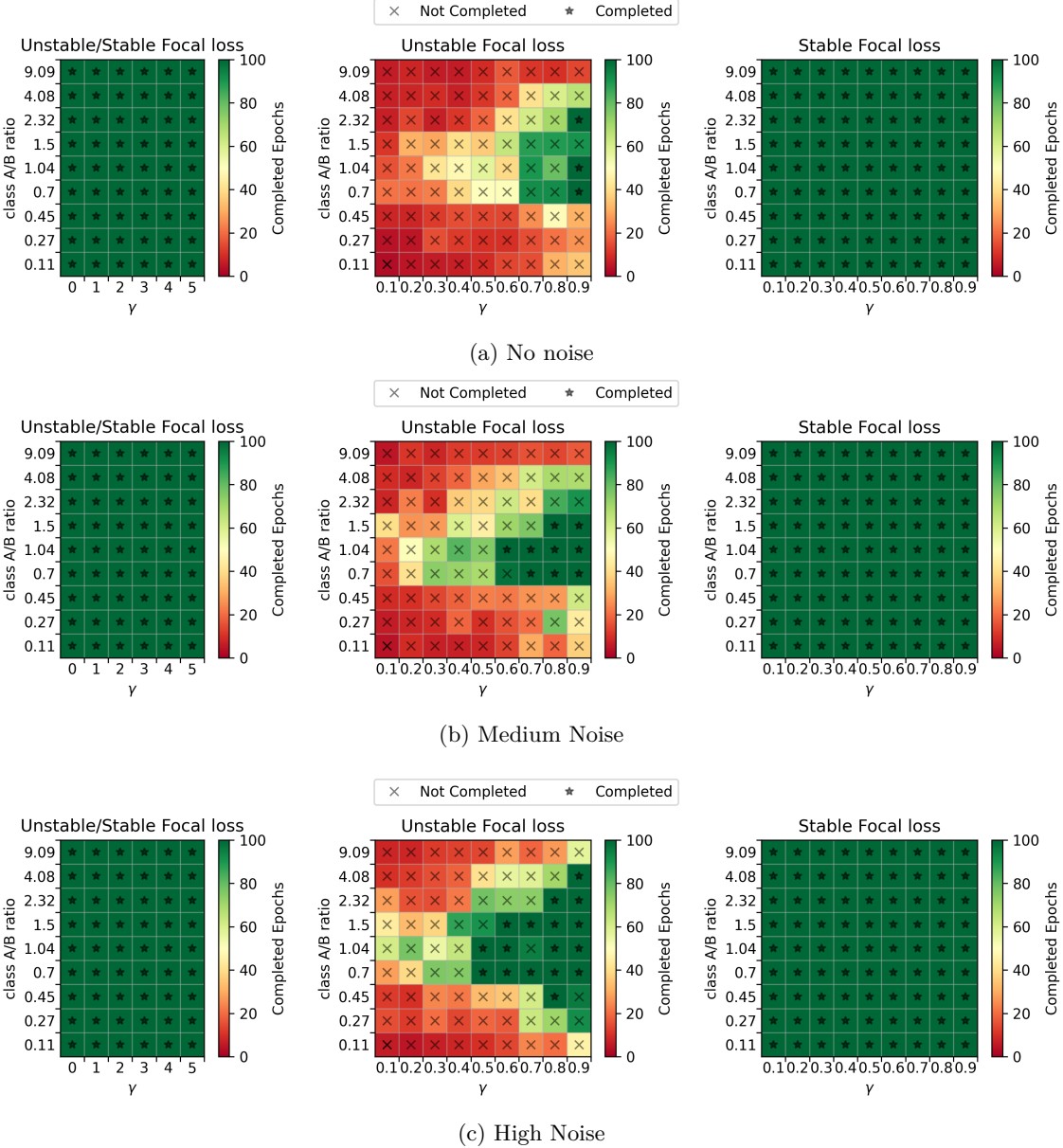

(a) No noise

(b) Medium Noise

(c) High Noise

Figure 4: ($a$): Binary classification results on the MNIST dataset, illustrating the number of completed epochs for different $\gamma$ values under varying class A/B ratios. The left plots show model results using $\gamma$ values of 0,1,2,3,4, and 5. The middle and left plots show the number of completed epochs whenever the $\gamma$ values of 0.1 to 0.9 with increments of 0.1 are used, where the middle plot shows the results when training with the original Focal Loss, and the plot on the right shows the results when using the stabilized Focal Loss. Each plot in the figures includes a marker that indicates whether all 100 epochs were completed. ($b$-$c$): Experiment results when the initial experiment is repeated with "Medium" and "High" amounts of noise added to the input images to increase the difficulty of the classification task.

which the results are shown on the right in Figure 4. These results show that the modification of the Focal Loss successfully eliminated the instability, as no more instabilities are reported in any of the experiments.

## 3.2 Binary Classification - CIFAR-10

Figure 5a shows the Focal Loss values when training the CNN and ViT to perform the binary classification tasks on the CIFAR-10 dataset. Similar to the classification results on the MNIST dataset, Figure 5a shows that the numerical instability occurs during training for both models. Figure 5b shows the results when the models were trained with the stabilized Focal Loss. Similar to the MNIST classification experiments, no further instabilities were reported when the models were trained with the stabilized Focal Loss, and both models completed training of all 1000 epochs.

## 3.3 2D Segmentation - MNIST

The results of training the 2D U-net are shown in Figure 6. This figure reports the computed Focal Loss for each epoch, but halted training whenever a "NaN" was encountered or all 1000 epochs were completed. When

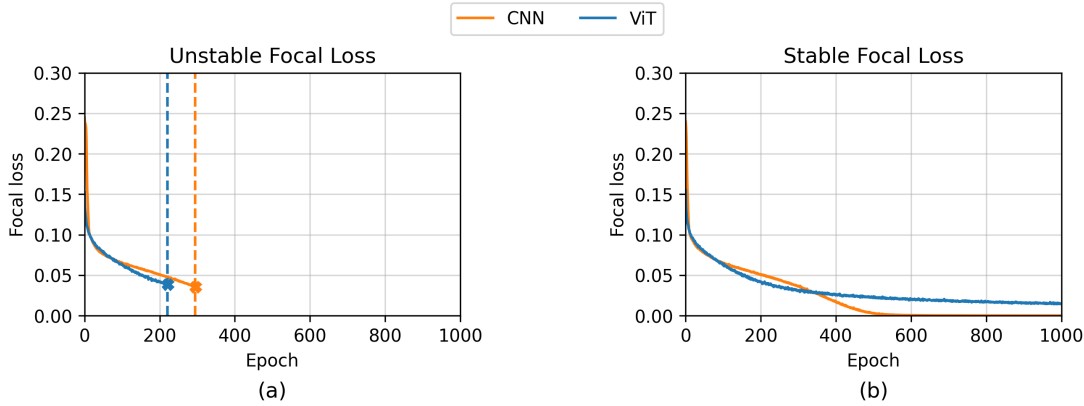

Figure 5: Results after training the CNN and ViT with the Focal Loss using a $\gamma$ and $\alpha_t$ of 0.5 for 1000 epochs on the CIFAR-10 dataset with the original Focal Loss ($a$), and the stabilized Focal Loss ($b$). A cross indicates the epoch at which a "NaN" was encountered during training.

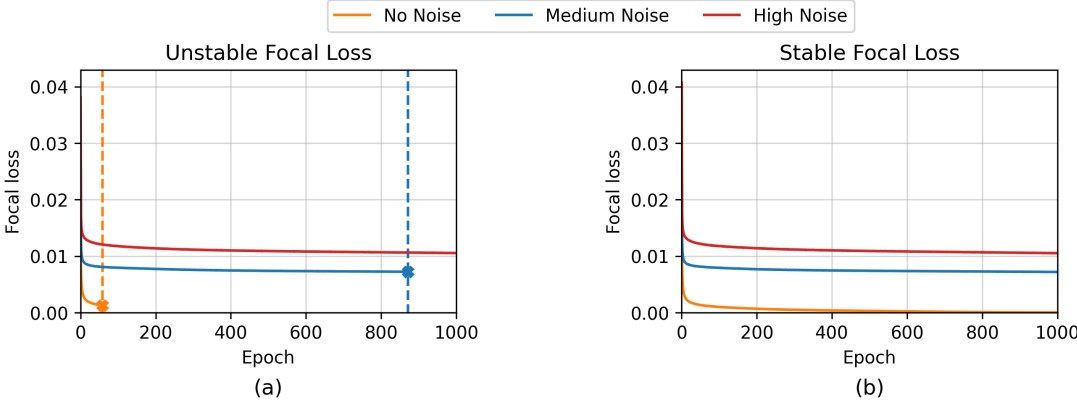

Figure 6: Results after training the 2D U-net with the Focal Loss using a $\gamma$ and $\alpha_t$ of 0.5 for 1000 epochs on the MNIST dataset with the original Focal Loss ($a$), and the stabilized Focal Loss ($b$). The model was trained without noise, and with "Medium" and "High" noise levels. A cross indicates the epoch at which a "NaN" was encountered during training.

no noise was added to the input data, instability was quickly observed. Introducing a small amount of noise delayed its onset, and at higher noise levels, instability was no longer observed. The segmentation results are consistent with the results from the binary classification task. Again, when using the modified version of the Focal Loss, no more instabilities were reported, and all three models were trained to completion. Note that the losses have different asymptotes, which originate from the way that the noise was added and the segmentation masks were generated. Introducing noise before generating the masks may have reduced the clarity of the boundary between foreground and background. Consequently, accurately determining this boundary becomes more difficult, leading to an increase in loss and a corresponding shift in the asymptote

## 4 Discussion and Conclusion

This paper addresses a hitherto unreported instability of the Focal Loss when the focussing parameter $\gamma$ is set to a value between 0 and 1. We showed that this instability is not only mathematically derivable but can also be demonstrated using some simple deep learning experiments. Due to the singularity that arises in the derivative of the Focal Loss when using these unstable $\gamma$ values, training deep learning models like a basic CNN, a ViT, or a 2D U-net can lead to unstable behavior. Our experiments suggest that when training with the Focal Loss using unstable $\gamma$ values, datasets with severe class imbalance are more prone to exhibiting instabilities earlier than when using balanced datasets, and that task complexity further influences the rate at which these instabilities manifest. A likely explanation is that models trained on easy tasks tend to overfit quickly, producing highly confident predictions. As a result, the model outputs become equal to the true class values, which leads to a singularity in the derivative of the Focal Loss consequently causing instability. The more difficult the task at hand, the more epochs are needed to reach a state where the predictions are confident enough to trigger the instability. This is also demonstrated in our experiments, which show that increasing task difficulty by adding noise delays the onset of the instability. With the presented experiments, we highlight that the instability is not necessarily an issue in all training scenarios, but that it can arise under certain conditions.

To resolve this instability, we proposed a modification of the original Focal Loss by adding a smoothing constant to the term that downscales the cross-entropy loss. This ensures that the singularity in the Focal Loss derivative is eliminated, consequently stabilizing model training. While unstable $\gamma$ values triggered instability with the original Focal Loss in our experiments, training with the modified version led to completing all epochs in each experiment.

In this paper, we provide numerical and experimental evidence of the existence of this Focal Loss instability. Our experiments highlight that the instability can be induced when training deep learning models. We therefore recommend refraining from using $\gamma$ values that fall between 0 and 1 when using the original Focal Loss. If by design, the only possible values for $\gamma$ fall between 0 and 1, as is the case for the Unified Focal Loss (Yeung et al., 2022), we recommend using our stabilized version of the Focal Loss to eliminate the chance of encountering instability. Yeung et al. (2022) did not report that their method suffered from numerical instabilities, even though their loss makes use of these unstable $\gamma$ values. Their published code shows clipping of the model outputs, which would prevent their model from reaching model outputs that trigger the instability. However, no explanations were provided for the clipping operation. Additionally, their paper focuses on complex segmentation tasks, which, as we have shown, are less prone to expressing instability. It could be possible that applying the Unified Focal Loss to more simplistic segmentation tasks could still trigger the instability.

In summary, we identified an unaddressed numerical instability in the Focal Loss and proposed the addition of a stabilizing smoothing constant to prevent it from destabilizing model training. Our experiments showed that after the addition of this smoothing constant to the Focal Loss, the instabilities were effectively removed. We therefore recommend either refraining from using the unstable $\gamma$ values when using the Focal Loss or adopting our modification to prevent instabilities from occurring.

## Acknowledgements

This research has been funded by a NWO Starter Grant (Project Reference: 510.023.062). Additionally, KVH gratefully acknowledges funding from the Dutch Research Council (NWO, Vidi grant no. 09150171910043)

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

## A  Appendix

### A.1  Derivation Focal Loss Derivative

This Appendix contains the derivations of the Focal loss derivative. We provide separate derivations for the foreground and background classes.

#### A.1.1  Complete Focal loss

$$\mathcal{L}_{\mathrm{F}}(y, p, \gamma, \alpha_t) = -\underbrace{\alpha_t y \,(1-p)^\gamma \log(p)}_{\mathcal{L}_{fg}} - \underbrace{(1-\alpha_t)(1-y)\, p^\gamma \log(1-p)}_{\mathcal{L}_{bg}} \tag{18}$$

#### A.1.2  Foreground Derivative

$$
\begin{aligned}
\frac{d\mathcal{L}_{fg}(p, \gamma, \alpha_t)}{dp}\Big|_{y=1} &= -\alpha_t \left( \frac{d(1-p)^\gamma}{dp} \log(p) + \frac{d\log(p)}{dp}(1-p)^\gamma \right) \\
&= -\alpha_t \left( -\gamma(1-p)^{\gamma-1} \log(p) + \frac{(1-p)^\gamma}{p} \right) \\
&= \alpha_t \left( \gamma(1-p)^{\gamma-1} \log(p) - \frac{(1-p)^\gamma}{p} \right)
\end{aligned}
\tag{19}
$$

### A.2  Background Derivative

$$
\begin{aligned}
\frac{d\mathcal{L}_{bg}(p, \gamma, \alpha_t)}{dp}\Big|_{y=0} &= -(1-\alpha_t) \left( \frac{dp^\gamma}{dp} \log(1-p) + \frac{d\log(1-p)}{dp}p^\gamma \right) \\
&= -(1-\alpha_t) \left( \gamma p^{\gamma-1} \log(1-p) - \frac{p^\gamma}{1-p} \right)
\end{aligned}
\tag{20}
$$

### A.3 Modified Focal Loss

In this Appendix, we show the Python implementation of the modified version of the original Focal loss Lin et al. (2017). Modifications to the original code are indicated by the "#Modification" comment.

```python
import torch
import torch.nn.functional as F

from torchvision.utils import _log_api_usage_once    #Modification

def sigmoid_focal_loss_modified(
    inputs: torch.Tensor,
    targets: torch.Tensor,
    alpha: float = 0.25,
    gamma: float = 2,
    reduction: str = "none",
    epsilon=1e-3 #Modification

) -> torch.Tensor:
    """
    Modified version of the Focal Loss. The epsilon scalar that is
    added to the output stabilizes the model training. Whenever
    epsilon is set to 0, it simplifies to the original Focal loss.

    Args:
        inputs (Tensor): A float tensor of arbitrary shape.
                The predictions for each example.
        targets (Tensor): A float tensor with the same shape as inputs.
                Stores the binary classification label for each element
                in inputs (0 for the negative class and
                1 for the positive class).
        alpha (float): Weighting factor in range (0,1) to balance
                positive vs negative examples or -1 for
                ignore. Default: ``0.25``.
        gamma (float): Exponent of the modulating factor (1 - p_t) to
                balance easy vs hard examples. Default: ``2``.
        epsilon(float): Smoothing constant preventing the
                instabilities when gamma values between 0 and 1
                are used. Default: ``1e-3``
        reduction (string): ``'none'`` | ``'mean'`` | ``'sum'``
                ``'none'``: No reduction will be applied to the output.
                ``'mean'``: The output will be averaged.
                ``'sum'``: The output will be summed.
                Default: ``'none'``.
    Returns:
        Loss tensor with the reduction option applied.
    """
    #Modification of the Original implementation from
    https://github.com/facebookresearch/fvcore/blob/master/fvcore/nn/focal_loss.py

    if not torch.jit.is_scripting() and not torch.jit.is_tracing():
        _log_api_usage_once(sigmoid_focal_loss_modified)  #Modification
    p = torch.sigmoid(inputs)
    ce_loss = F.binary_cross_entropy_with_logits(inputs, targets,
                reduction="none")
    p_t = (p) * targets + (1 - p) * (1 - targets)
    loss = ce_loss * ((1 - p_t+epsilon) ** gamma) #Modification

    if alpha >= 0:
        alpha_t = alpha * targets + (1 - alpha) * (1 - targets)
        loss = alpha_t * loss

    # Check reduction option and return loss accordingly
    if reduction == "none":
        pass
```

```python
63        elif reduction == "mean":
64            loss = loss.mean()
65        elif reduction == "sum":
66            loss = loss.sum()
67        else:
68            raise ValueError(
69                f"Invalid Value for arg 'reduction': '{reduction} \n
70                Supported reduction modes: 'none', 'mean', 'sum'"
71            )
72        return loss
73
```

### A.4 CNN for Binary and Multiclass classification

A summary of the CNN (modified from a CNN in a Pytorch Tutorial `https://pytorch.org/tutorials/beginner/blitz/cifar10_tutorial.html`) that was used for binary classification is shown in Table 3. A code snippet that displays the implementation of this model in Python is also provided below.

Table 2: Summary of the binary classification CNN when using a batch size of 64 to train on the MNIST dataset

| Layer (type:depth-idx) | Output Shape | Param # |
|---|---|---|
| CNN | [64, 1] | – |
| Conv2d: 1-1 | [64, 6, 24, 24] | 156 |
| MaxPool2d: 1-2 | [64, 6, 12, 12] | – |
| Conv2d: 1-3 | [64, 16, 8, 8] | 2.416 |
| MaxPool2d: 1-4 | [64, 16, 4, 4] | – |
| Linear: 1-5 | [64, 120] | 30.840 |
| Linear: 1-6 | [64, 84] | 10.164 |
| Linear: 1-7 | [64, 1] | 85 |

Table 3: Summary of the binary classification CNN when using a batch size of 128 to train on the CIFAR-10 dataset

| Layer (type:depth-idx) | Output Shape | Param # |
|---|---|---|
| CNN | [128, 3] | – |
| Conv2d: 1-1 | [128, 6, 28, 28] | 156 |
| MaxPool2d: 1-2 | [128, 6, 14, 14] | – |
| Conv2d: 1-3 | [128, 16, 10, 10] | 2.416 |
| MaxPool2d: 1-4 | [128, 16, 5, 5] | – |
| Linear: 1-5 | [128, 120] | 48.120 |
| Linear: 1-6 | [128, 84] | 10.164 |
| Linear: 1-7 | [128, 1] | 85 |

```python
import torch.nn as nn
import torch.nn.functional as F
import torch
from torchinfo import summary

class CNN(nn.Module):
    def __init__(self,no_classes=1):
        super().__init__()
        self.channels=channels
        Conv_output_size=int((((input_size-4)/2)-4)/2)
        self.conv1 = nn.Conv2d(channels, 6, 5)
        self.pool = nn.MaxPool2d(2, 2)
        self.conv2 = nn.Conv2d(6, 16, 5)
        self.fc1 = nn.Linear(Conv_output_size*Conv_output_size*16, 120)
        self.fc2 = nn.Linear(120, 84)
        self.fc3 = nn.Linear(84, no_classes)

    def forward(self, x):
        x = self.pool(F.relu(self.conv1(x)))
        x = self.pool(F.relu(self.conv2(x)))
        x = torch.flatten(x, 1)

        x = F.relu(self.fc1(x))
        x = F.relu(self.fc2(x))
        x = self.fc3(x)
        return x
```

**A.5   Architecture VisionTransformer (ViT)**

Summary of the Vision Transformer architecture. More details on this architecture can be found at `https://huggingface.co/google/vit-base-patch16-224`.

Table 4: Summary of the Vision Transformer architecture

| Layer (type:depth-idx) | Output Shape | Param # |
|---|---|---|
| VisionTransformer | – | 152,064 |
| PatchEmbed: 1-1 | – | – |
|   Conv2d: 2-1 | – | 590,592 |
|   Identity: 2-2 | – | – |
| Dropout: 1-2 | – | – |
| Identity: 1-3 | – | – |
| Identity: 1-4 | – | – |
| Sequential: 1-5 | – | – |
|   Block: 2-x (repeated 14 times) | – | – |
|     LayerNorm | – | 1,536 |
|     Attention | – | 2,362,368 |
|     Identity | – | – |
|     Identity | – | – |
|     LayerNorm | – | 1,536 |
|     Mlp | – | 4,722,432 |
|     Identity | – | – |
|     Identity | – | – |
| LayerNorm: 1-6 | – | 1,536 |
| Identity: 1-7 | – | – |
| Dropout: 1-8 | – | – |
| Linear: 1-9 | – | 769 |
| Total params: | | 85,799,425 |
| Trainable params: | | 85,799,425 |

### A.6 U-Net

The U-Net Ronneberger et al. (2015) implementation used in this paper was adapted from `https://github.com/clemkoa/u-net/blob/master/unet/unet.py` with some minor modifications.

Table 5: Summary of the 2D U-Net architecture when using a batch size of 64

| Layer (type:depth-idx) | Output Shape | Param # |
|---|---|---|
| UNet | [64, 1, 28, 28] | – |
| Conv2d: 3-1 | [64, 64, 28, 28] | 640 |
| BatchNorm2d: 3-2 | [64, 64, 28, 28] | 128 |
| Conv2d: 3-4 | [64, 64, 28, 28] | 36,928 |
| BatchNorm2d: 3-5 | [64, 64, 28, 28] | 128 |
| Sequential: 3-7 | [64, 128, 14, 14] | 221,952 |
| Sequential: 3-8 | [64, 256, 7, 7] | 886,272 |
| Sequential: 3-9 | [64, 512, 3, 3] | 3,542,016 |
| Sequential: 3-10 | [64, 1024, 1, 1] | 14,161,920 |
| ConvTranspose2d: 3-11 | [64, 512, 2, 2] | 2,097,664 |
| Sequential: 3-12 | [64, 512, 3, 3] | 7,080,960 |
| ConvTranspose2d: 3-13 | [64, 256, 6, 6] | 524,544 |
| Sequential: 3-14 | [64, 256, 7, 7] | 1,771,008 |
| ConvTranspose2d: 3-15 | [64, 128, 14, 14] | 131,200 |
| Sequential: 3-16 | [64, 128, 14, 14] | 443,136 |
| ConvTranspose2d: 3-17 | [64, 64, 28, 28] | 32,832 |
| Sequential: 3-18 | [64, 64, 28, 28] | 110,976 |
| Conv2d: 1-10 | [64, 1, 28, 28] | 65 |
| **Total params** | | 31,042,369 |
| **Trainable params** | | 31,042,369 |
| **Non-trainable params** | | 0 |
| **Total mult-adds (G)** | | 36.16 |
| **Input size (MB)** | | 0.20 |
| **Forward/backward size (MB)** | | 425.34 |
| **Params size (MB)** | | 124.17 |
| **Estimated Total Size (MB)** | | 549.71 |

```python
#Downloaded and modified from:
#https://github.com/clemkoa/u-net/blob/master/unet/unet.py

import torch
from torch import nn
import torch.nn.functional as F

class DoubleConv(nn.Module):
    def __init__(self, in_ch, out_ch):
        super(DoubleConv, self).__init__()
        self.conv = nn.Sequential(
            nn.Conv2d(in_ch, out_ch, kernel_size=3, padding=1),
            nn.BatchNorm2d(out_ch),
            nn.ReLU(inplace=True),
            nn.Conv2d(out_ch, out_ch, kernel_size=3, padding=1),
            nn.BatchNorm2d(out_ch),
            nn.ReLU(inplace=True),
        )

    def forward(self, x):
        x = self.conv(x)
        return x

class Up(nn.Module):
    def __init__(self, in_ch, out_ch):
        super(Up, self).__init__()
        self.up_scale = nn.ConvTranspose2d(in_ch, out_ch,
        kernel_size=2, stride=2)

    def forward(self, x1, x2):
        x2 = self.up_scale(x2)

        diffY = x1.size()[2] - x2.size()[2]
```

```python
34          diffX = x1.size()[3] - x2.size()[3]
35
36          x2 = F.pad(x2, [diffX // 2, diffX - diffX // 2,
37          diffY // 2, diffY - diffY // 2])
38          x = torch.cat([x2, x1], dim=1)
39          return x
40
41
42 class DownLayer(nn.Module):
43     def __init__(self, in_ch, out_ch):
44         super(DownLayer, self).__init__()
45         self.pool = nn.MaxPool2d(2, stride=2, padding=0)
46         self.conv = DoubleConv(in_ch, out_ch)
47
48     def forward(self, x):
49         x = self.conv(self.pool(x))
50         return x
51
52
53 class UpLayer(nn.Module):
54     def __init__(self, in_ch, out_ch):
55         super(UpLayer, self).__init__()
56         self.up = Up(in_ch, out_ch)
57         self.conv = DoubleConv(in_ch, out_ch)
58
59     def forward(self, x1, x2):
60         a = self.up(x1, x2)
61         x = self.conv(a)
62         return x
63
64
65 class UNet(nn.Module):
66     def __init__(self,channels=1, dimensions=1):
67         super(UNet, self).__init__()
68         self.conv1 = DoubleConv(channels, 64)
69         self.down1 = DownLayer(64, 128)
70         self.down2 = DownLayer(128, 256)
71         self.down3 = DownLayer(256, 512)
72         self.down4 = DownLayer(512, 1024)
73         self.up1 = UpLayer(1024, 512)
74         self.up2 = UpLayer(512, 256)
75         self.up3 = UpLayer(256, 128)
76         self.up4 = UpLayer(128, 64)
77         self.last_conv = nn.Conv2d(64, dimensions, 1)
78
79     def forward(self, x):
80         x1 = self.conv1(x)
81         x2 = self.down1(x1)
82         x3 = self.down2(x2)
83         x4 = self.down3(x3)
84         x5 = self.down4(x4)
85         x1_up = self.up1(x4, x5)
86         x2_up = self.up2(x3, x1_up)
87         x3_up = self.up3(x2, x2_up)
88         x4_up = self.up4(x1, x3_up)
89         output = self.last_conv(x4_up)
90         return output
91
```

