# OpenReview forum: "A Note On The Stability Of The Focal Loss"
_TMLR — Accepted by TMLR_

### Review · Reviewer_EoP2 · 2025-07-14

**Summary Of Contributions:**

The authors investigate a feature of the focal loss where for gamma parameter values between 0 and 1, the focal loss becomes unstable. Theoretically, the authors show that this instability occurs when the model output is very close to correct, which results in a division by zero in the computation of the model output gradient. The authors demonstrate this effect empirically and also propose a novel modification to the focal loss to avoid this instability.

**Audience:**

Yes

**Broader Impact Concerns:**

No broader impact concerns.

**Claims And Evidence:**

Yes

**Requested Changes:**

**Critical**

Please include experiments on a larger dataset (such as CIFAR or ideally ImageNet)

Please modify Figures 1 and 2 as described above

**Strengths And Weaknesses:**

**Strengths**

The authors consider a very specific and well defined problem, and do overall do a good job at both defining and addressing it. The theoretical analysis is sound and is well-described. Empirically, the authors consider both classification and segmentation experiments, which is good practice. This paper would likely have signifiance to practitions who frequently use the focal loss.

**Weaknessess**

The primary weakness in my view is that the experiments are quite small scale. It's difficult to assess whether the results would extend beyond a simple setting like MNIST. I would encourage the authors to at least experiment on CIFAR-10 or ideally an ImageNet-scale dataset, although I understand that some careful tuning may be required to observe this instability on larger datasets. This would help establish the empirical significance of this work.

A more minor concern is on Figures 1 and 2. These figures aims to show a qualitivative difference in the gradients between a gamma value of 0.5 and the other gamma values, which is central to the argument of why focal loss fails in this particular regime. Unfortunately, neither figure explicitly depicts a discontinuity at the point where instability ocurrs (presumably p=1.0 in figure 1c for example). For Figure 1cd, I would encourage including an empty circle or an x to indicate when the gradient is undefined for the orange curve. For Figure 2, I would similarly indicate when the gradient is undefined: all this figure appears to show now is when the gradients are undefined for the opposite reason to what the authors consider (namely, when the model is highly incorrect).

---

> ### Author Response · Authors · 2025-08-29
> **Response to Reviews**
>
> Dear reviewer, thank you for taking the time to review our submission to TMLR and overall positive assessment of our work.
>
>
> As to the first concern, we agree with the reviewer’s suggestion to additionally showcase the instability on a more complex dataset and have therefore performed additional classification experiments on a more complex and realistic dataset, the CIFAR-10 dataset, as recommended. In our experiments, we divided the CIFAR-10 dataset into 2 classes (animals vs vehicles), leading to a 60:40 class A/B ratio, and trained two classifiers: a CNN and a Vision Transformer (ViT). The models were set to train with a \gamma of 0.5 for 1000 epochs. Our results showed that the instability was detected after approximately 200 epochs for the ViT and 300 for the CNN classifier.  With our modification of the Focal loss, both models completed training up to 1000 epochs.  A more detailed description of this additional experiment can be found in subsection 2.5.2  and subsection 3.2 on pages 7 and 9 of the revised manuscript.
>
>
> Regarding the second point, we agree with the reviewer that it would be useful to indicate when the gradients become undefined in Figure 1c and Figure 1d. We thank the reviewer for this insightful suggestion and have modified the figure in the revised manuscript accordingly. For Figure 2c and Figure 2d, we would like to clarify that the points that are indicated in the figures do, in fact, indicate the values of $x$ that are equal to the ground truth label. For clarification, we have added the following sentence in the caption of this figure:
>
>
> “ The orange marker represents the model output $x$  that is approximately equal to 1 in Figure (c) and equal to 0 in Figure (d)."

---

> > ### Comment · Reviewer_EoP2 · 2025-09-05
> >
> > Thank you for your response! I'm satisfied by the additional experiments and clarification regarding the figures.

---

### Review · Reviewer_Erbc · 2025-07-27

**Summary Of Contributions:**

This paper identifies and analyzes the instability of the Focal Loss function. The authors demonstrate that this instability arises when the focusing parameter $\gamma$ is set between 0 and 1 (0.5 is used as the example), which can lead to numerical issues during model training. Specifically, the denominator in the loss function may approach zero as the predicted probability converges to the true label. To address this, the authors propose a simple yet effective modification to the Focal Loss by introducing a small constant $\epsilon$, a common technique in numerical computation to avoid 0 denominator. Experimental results on binary classification and segmentation tasks using the MNIST dataset demonstrate the effectiveness of the proposed method.

**Audience:**

Yes

**Broader Impact Concerns:**

I have not found any discussions about the limitations and potential negative societal impact. But in my opinion, this may not be a problem, since the work only focuses on the optimization in deep learning. Still, it is highly encouraged to add corresponding discussions.

**Claims And Evidence:**

Yes

**Requested Changes:**

See weakness

**Strengths And Weaknesses:**

**Strengths**
1. The paper is clearly written and easy to follow.
2. Training to stabilize focal loss is an area of interest in optimization.
3. The authors identifies and analyses the instability of the Focal Loss function from the perspectives of numerical instability, which for me is a quite interesting perspective.
4. The proposed method is simple and effective, which can be easily implemented in practice.

** Weakness **
1. This instability is caused by the denominator approaches 0 when predicted probability converges to the true label. While the proposed solution (adding a small constant to the denominator to prevent division by zero) is indeed effective, it is a well-established technique in numerical computation for addressing singularities. From this aspect, the contribution may be viewed as more of a practical fix than an academic advancement.

2. The authors have only demonstrate the instability with $\gamma = 0.5$. The authors could shape such a conclusion via a more rigorous theoretical analysis.

3. No related works are discussed. Several existing studies may also impact the stability of Focal Loss, and the authors should analyze and compare their method with such works. Relevant references include:

    [1] Islam, Md Rakibul, et al. "Enhancing Semantic Segmentation with Adaptive Focal Loss: A Novel Approach." arXiv preprint arXiv:2407.09828 (2024).

    [2] Li, Xiang, et al. "Generalized focal loss: Learning qualified and distributed bounding boxes for dense object detection." Advances in neural information processing systems 33 (2020): 21002-21012.


4. The empirical results are not sufficient to support the effectiveness of the proposed method. The authors only provide results on MNIST dataset, which is a very simple dataset. For now, this is not sufficiently comprehensive for the current standards within the deep learning community. The authors should provide more comprehensive experiments on more complex datasets to demonstrate the effectiveness of the proposed method.

---

> ### Author Response · Authors · 2025-08-29
> **Response to Review**
>
> We would like to thank the reviewer for their time and insightful comments. Below, we address each comment and demonstrate the changes we will make to the manuscript.
>
>
> 1. Although we do agree with the reviewer that our proposed solution can be viewed as a practical fix, we would like to emphasize its academic value. Our contribution highlights the importance of carefully considering potential weaknesses that a new loss function can exhibit, to prevent undesired instabilities from occurring.
>
>
> 2. Even though it is true that in our methods section we show an example of the instability whenever a $\gamma$ of 0.5 is used, the instability occurs for all other $\gamma$ values that fall between 0 and 1 (as mentioned in the last sentence of the second paragraph on page 3: “ Although in this example γ was set to 0.5, this holds for all γ values between 0 and 1, as all these values introduce a fraction in the derivative with the model output in the denominator”. The occurrence of the instability for all values of $\gamma$ within the unit interval is also demonstrated in our binary classification experiments. For the segmentation and the CIFAR-10 binary classification task, we performed the analysis with $\gamma = 0.5$, and showed that also in this scenario the instability was detected.
>
>
> 3. We thank the reviewer for suggesting the other studies that are related to the focal loss. We have updated our paper by mentioning these works and by describing how they relate to our work by adding a separate paragraph to the introduction (fourth paragraph of the revised manuscript’s introduction on page 2.)
>
>
> 4. We chose to demonstrate our work on the MNIST dataset because of its simplicity, making it easy to run multiple experiments while being able to change the difficulty of our classification and segmentation task easily. Nevertheless, we agree with the reviewer that this is a simple dataset, and we thank the reviewer for the suggestion to extend the experiments. We ran a binary classification experiment on the CIFAR-10 dataset using the previously used CNN and a vision transformer. We have been able to detect the instability in these experiments and would like to refer the reviewer to our response to Reviewer 1 for more details.
> As to the reviewer’s comment regarding discussing our work’s potential societal impact: We refrain from discussing any societal impact, as we are addressing a numerical instability, making it difficult to determine what the societal impact is. However, the additional paragraph that has been added (as mentioned in point 3) showcases that by addressing the instability, we can potentially also stabilize other loss functions that are based on the Focal loss. Therefore, the impact that this paper has is not exclusively on the Focal loss, but rather all Focal loss-inspired loss functions.

---

### Review · Reviewer_Zb67 · 2025-08-23

**Summary Of Contributions:**

The paper first attempts to show that their is a mathematic instability of the gradients of the focal loss when $\gamma$ is between 0 and 1.
This results in a divide by $p$ raised to a fractional power and thus the possibility of divide by 0.
The paper claims (incorrectly) that in the limit the gradients become undefined when the classifier is super confident.
The paper proposes to add an positive epsilon to the denominator to avoid the divide by zero error.
Then the paper showcases the instability in a simple MNIST experiment where the class imbalance and the difficulty of the problem is carefully controlled.

**Audience:**

Yes

**Claims And Evidence:**

No

**Requested Changes:**

- Ultimately, I believe the instability observed is NOT a mathematic instability but rather a numerical instability. Thus, the fix should not be to modify the loss function but to compute the loss function in a numerically stable way. This should be the focus of the paper, not modifying the loss function. Please look into numerically stable ways to compute the focal loss function.

- Also, you need to completely rewrite the story if this is only a numerical instability, which I believe it is. The problem only showcases itself for extremely small values close to 0 or 1.

- (Less critical but still important) Showcase a more realistic example where this instability causes problems. Without this, the motivation or relevance to the ML audience is questionable. I think you could possibly show it with a very small $\gamma$ like 0.01 or 0.0001 even on a somewhat more realistic dataset. For example, the experiments in Unified Focal Loss that used clipping. If you can show that their results do not hold without clipping, then you have a strong case.

- (Ideally) You would show that the numerically stable version is within floating-point precision.

**Strengths And Weaknesses:**

**Strengths**

- The paper has a simple toy experiment to showcase the instability in practice and shows that the "difficulty" of the problem is critical. It only shows up in "easy" problems.

- The paper's flow is relatitvely easy to follow.


**Weaknesses**
- (Major and fundamental) Incorrect derivation that is fundamental to the paper's claims. The paper claims in (8) and (9) that there is a division by zero in the limit. But this is not true (also, there seems to be notational inconsistency since the limit is for $\hat{y}$ but I'm assuming it is really for $p$. The limit needs to use L'hopitals rule since both the numerator  $\log(1-p)$ and denominator $\sqrt{p}$ approach 0 as $p$ approaches 0. Taking the derivative of both we have $\frac{1}{1-p}$ and $\frac{1}{\sqrt{p}}$ so the fraction becomes $\frac{\sqrt{p}}{1-p}$, which then simplifies to simply 0 instead of $\frac{0}{0}$ as is claimed.


- It is unclear if this is actually a practical problem or not given the problem above. I'm not convinced this is even a fundamental problem. Perhaps, it is a numerical instability problem but it should be addressed by doing numerically stable calculations rather than by modifying the Focal loss function itself to be different.k

- The experiments are very simple and may suggest that indeed this numerical instability does not happen in practice since you have to have very extreme values before it causes numerical issues (38 or -38 is super high and probably just a numerical issue, not a fundamental instability).

---

> ### Author Response · Authors · 2025-08-29
> **Response to Review (1/2)**
>
> We are very grateful to the reviewer for pointing out a mistake in our derivation of the limit of the Focal Loss derivative. Below, we address the comments and highlight the changes made to the manuscript.
>
> 1. Thank you for the valuable feedback on the incorrect derivation of the limit of the Focal loss derivative.  We agree that mathematically, these limits approach zero based on L’Hôpital’s rule, as opposed to the limits being undefined. The reviewer is therefore correct that, from a theoretical point of view, as long as the model outputs ($p$) do not reach the actual ground truth value, the Focal loss derivative cannot become undefined. As such, we fully agree with the reviewer that the observed instability is not mathematical instability, but that it must be a numerical instability.
> Indeed, as we critically reevaluated our original derivations, we additionally realised that using limit equations to make our point was inappropriate in the first place. Rather, we ought to have considered a case where the exact ground truth labels are being predicted. These model outputs can be reached during the numerical computation of the loss as a result of the limited precision of floating-point numbers. To clarify that we are specifically focusing on a case in which the model output becomes equal to the ground truth, we have therefore replaced the limit equations (Equations 8,9,16, and 17) with equations that show that the derivative becomes undefined whenever the model outputs become equal to the ground truth labels. In addition, we have extended paragraph 2.3 so that it is now clear that the instability arises as a result of a numerical rather than a mathematical instability.
>
>  2. We thank the reviewer for suggesting we rethink the story of the manuscript, given that we address a numerical instability as opposed to a mathematical instability. However, we believe that the story that we are portraying in this manuscript is still appropriate the way we have presented it.
> The goal of this manuscript was not to address a limitation of the mathematical/analytical definition of the Focal loss but rather to point out an unaddressed instability that can be encountered during training of a machine/deep learning model when using this loss with specific $\gamma$ values. Because we wanted to provide insights into the origin of this instability, we made use of the mathematical derivation of the Focal loss and its derivative, in which we showcase that this instability only occurs when a $\gamma$ value between 0 and 1 is used and model output values close to the ground truth label are produced. This is still the case, albeit that the (incorrect) derivation of the limit has now been replaced by a derivation of the derivative whenever the model outputs are equal (within numerical precision) to the ground truth labels.
> The experiments we included in the manuscript are meant to illustrate that the instability can be encountered during training of a machine learning model when using these unstable $\gamma$ values, and that it is not encountered when using the stable $\gamma$ values. These results can be explained using the mathematical equations of the Focal loss and its derivative. We also still make use of the mathematical derivations to come up with a solution to the numerical instability that is encountered when using these unstable $\gamma$ values. We therefore want to emphasize that the included mathematical equations in the manuscript were necessary to provide a theoretical understanding of the numerical instability, rather than demonstrating that the Focal loss is mathematically unstable.
>
> To further ensure that it is absolutely clear that we address a numerical instability, as opposed to a mathematical one, we have made the following adaptations throughout the manuscript:
>
> •	Change “unaddressed instability" to "unaddressed numerical instability" throughout the paper.
>
> •	Emphasize that the instability is a numerical issue

---

> ### Author Response · Authors · 2025-08-29
> **Response to Review (2/2)**
>
> 3. We thank the reviewer for suggesting that we should expand our experiments to a more complex dataset. We have performed additional experiments to showcase the instability on a more complex dataset. In these experiments, we trained a binary classification model on the CIFAR-10 dataset using a $\gamma$ of 0.5. For more details on this experiment, we kindly refer you to the response of Reviewer 1.
>
>
> 4. We would like to thank the reviewer again for pointing out that the instability that we address in this paper is not a mathematical instability, but a numerical one. As mentioned in our first comment, we will make sure that throughout the paper, it is clear that we consider the instability that we address to be a numerical instability. We would like to emphasize that our modification (the addition of a smoothing parameter $\epsilon$) is not meant to alter the overall behaviour of the Focal loss but is only added to ensure numerical stability. The addition of a smoothing parameter to ensure numerical stability is a common approach in machine learning, as it is also done in the computation of other well-known loss functions, such as the Dice loss, in which a division by zero is also prevented by the usage of a smoothing parameter $\epsilon$. Similarly, the well-known batch normalization technique also makes use of a smoothing parameter $\epsilon$ to prevent division by zero. The addition of these smoothing parameters does not alter the behaviour of the functions. We therefore consider our “modification” to the Focal loss to be a conventional method to ensure numerical stability, which is in line with how this problem is dealt with throughout the machine learning community.

---

> > ### Comment · Reviewer_Zb67 · 2025-09-22
> >
> > Thanks for the comments. I appreciate the acknowledgement that it is a numerical instability issue. I still think the paper needs to be rewritten with this as the focus rather than merely modifying the current paper. I am also not fully convinced that this is a problem that happens in practice. A real-world example where this actually happens in practice (not just on specific benchmark datasets in particular settings) would have improved the paper.

---

### Comment · Action_Editor_Wn5N · 2025-08-23
**Author rebuttal**

Dear Authors,

Now with all the reviews available, authors shall post their response.

Thanks,

AC

---

### Decision · Action_Editor_Wn5N · 2025-10-03

**Recommendation:** Accept with minor revision

**Audience:**

Yes

**Audience Explanation:**

Since focal loss is widely used in practice, particularly in classification and object detection tasks with dataset imbalance. Ensuring stability in such commonly deployed losses is practically important, and even simple fixes can be impactful in large-scale real-world systems.

**Claims And Evidence:**

Yes

**Claims Explanation:**

The paper received mixed reviews but overall leans toward acceptance. A Reviewer raised concerns about whether the identified instability in focal loss is a genuine real-world problem and suggested it could be handled with standard numerically stable functions. Reviewers 2 and 3, however, are in favor of acceptance, noting that the main concerns were addressed in the rebuttal.

It is worth emphasizing that focal loss is widely used in practice, particularly in classification and object detection tasks with dataset imbalance. Ensuring stability in such commonly deployed losses is practically important, and even simple fixes can be impactful in large-scale real-world systems.

I recommend acceptance, but the authors should carefully address the concerns raised, particularly clarifying the practical contexts where the instability arises and why their solution is significant. Strengthening the exposition in the final version will enhance the paper’s clarity and impact.